# An Integrative Systems Biology and Experimental Approach Identifies Convergence of Epithelial Plasticity, Metabolism, and Autophagy to Promote Chemoresistance

**DOI:** 10.3390/jcm8020205

**Published:** 2019-02-07

**Authors:** Shengnan Xu, Kathryn E. Ware, Yuantong Ding, So Young Kim, Maya U. Sheth, Sneha Rao, Wesley Chan, Andrew J. Armstrong, William C. Eward, Mohit Kumar Jolly, Jason A. Somarelli

**Affiliations:** 1Duke Cancer Institute and the Department of Medicine, Duke University Medical Center, Durham, NC 27710, USA; shengnan.xu@duke.edu (S.X.); kathryn.ware@duke.edu (K.E.W.); maya.sheth@duke.edu (M.U.S.); wesley.chan@duke.edu (W.C.); andrew.armstrong@duke.edu (A.J.A.); 2Department of Biology, Duke University Medical Center, Durham, NC 27710, USA; yuantong.ding@duke.edu; 3Department of Molecular Genetics and Microbiology, Duke University Medical Center, Durham, NC 2 7710, USA; soyoung.kim@duke.edu; 4Department of Orthopaedic Surgery, Duke University Medical Center, Durham, NC, 27710, USA; sneha.rao@duke.edu (S.R.); william.eward@duke.edu (W.C.E.); 5Solid Tumor Program and the Duke Prostate and Urologic Cancer Center, Duke University Medical Center, Durham, NC 27710, USA; 6Center for Theoretical Biological Physics, Rice University, Houston, TX 77005-1827, USA; 7Current address: Centre for BioSystems Science and Engineering, Indian Institute of Science, Bangalore 560012, India

**Keywords:** evolution, systems biology, autophagy, lung cancer, epithelial–mesenchymal transition, tumor invasiveness, metabolism

## Abstract

The evolution of therapeutic resistance is a major cause of death for cancer patients. The development of therapy resistance is shaped by the ecological dynamics within the tumor microenvironment and the selective pressure of the host immune system. These selective forces often lead to evolutionary convergence on pathways or hallmarks that drive progression. Thus, a deeper understanding of the evolutionary convergences that occur could reveal vulnerabilities to treat therapy-resistant cancer. To this end, we combined phylogenetic clustering, systems biology analyses, and molecular experimentation to identify convergences in gene expression data onto common signaling pathways. We applied these methods to derive new insights about the networks at play during transforming growth factor-β (TGF-β)-mediated epithelial–mesenchymal transition in lung cancer. Phylogenetic analyses of gene expression data from TGF-β-treated cells revealed convergence of cells toward amine metabolic pathways and autophagy during TGF-β treatment. Knockdown of the autophagy regulatory, ATG16L1, re-sensitized lung cancer cells to cancer therapies following TGF-β-induced resistance, implicating autophagy as a TGF-β-mediated chemoresistance mechanism. In addition, high ATG16L expression was found to be a poor prognostic marker in multiple cancer types. These analyses reveal the usefulness of combining evolutionary and systems biology methods with experimental validation to illuminate new therapeutic vulnerabilities for cancer.

## 1. Introduction

Mammalian cells respond to external stimuli through a coordinated system of signaling and gene expression circuitry. The inputs to this system are often the ligands for receptors, which initiate signaling cascades that ultimately lead to changes in gene expression. A cell can receive, process, and integrate multiple simultaneous inputs and respond to them with a coordinated phenotypic response [1,2].

Deregulation of the cellular signaling/response circuitry is a fundamental theme in cancer at both the tissue and single-cell levels. Indeed, deregulated intracellular signaling/gene expression circuitry is fundamental to many cancer hallmarks [3], including sustaining proliferation [4,5], evading growth suppression [5], inducing angiogenesis [5], tumor-promoting inflammation [5], invasion [6], and metastasis [7,8,9].

One well-studied signaling/expression circuit that is frequently dysregulated in cancer is the transforming growth factor β (TGF-β)/SMAD axis. The TGF-β/SMAD axis is a critical developmental pathway that controls differentiation and proliferation [10]. TGF-β/SMAD signaling is also important in wound healing and fibrosis (reviewed in [11,12]). One of the major phenotypic outputs of TGF-β/SMAD signaling is the phenotypic switch from an epithelial to a mesenchymal state, known traditionally as epithelial–mesenchymal transition (EMT) (reviewed in [13]). In the context of cancer, TGF-β-mediated EMT promotes downregulation of cell–cell adhesion and upregulation of migration and invasion [14,15]. This pro-invasive phenotype is usually activated at the expense of proliferation [15,16]: TGF-β induces potent cell cycle arrest through SMAD-mediated transcriptional activation of the cell cycle repressor, p21 [17]. TGF-β also reprograms cellular metabolism [18] and induces autophagy [19]—a process in which a cell self-digests its proteins and organelles. In addition to its cell autonomous role in promoting invasiveness, TGF-β also acts non-cell autonomously to create a tumor microenvironment more permissive to tumor growth [20,21]. These mechanisms can often drive resistance to chemotherapy and multiple targeted therapies [22,23].

However, the abovementioned effects of TGF-β/SMAD-induced EMT are typically studied in isolation with focus on a few nodes of the pathway, thereby neglecting the effects of crosstalk among multiple signaling pathways. Such crosstalk can often generate feedback loops with nonlinear dynamics, giving rise to emergent, complex, and non-intuitive behavior [24]. Hence, a systems biology approach, integrating computational and experimental components, can be essential to elucidating the dynamics of underlying interconnected cellular circuitry and identifying the fundamental organizational principles driving tumor progression [25]. Here, we used such an approach, incorporating multiple systems biology tools to analyze the dynamics of TGF-β-mediated EMT and to experimentally validate the computationally derived insights (Figure 1). 

Cancer progression is an evolutionary process of selection over time [26,27]. Therefore, we postulated that tools developed for tracing evolutionary histories may provide new insights. One of the most commonly used methods of inferring ancestral relationships is phylogenetics. Phylogenetics uses a data matrix of character states to infer evolutionary relationships between groups [28]. Although phylogenetics was originally developed to reconstruct ancestral relationships between species, phylogenetic inference has also been applied to diverse datasets for which no underlying ancestral relationships exist, such as geography, linguistics, or astrophysics [28].

Given the flexibility of phylogenetics as a clustering tool for multiple data types and contexts, we hypothesized that analysis of time course gene expression data could provide crucial information about how circuits are integrated to lead to a given phenotype. We identified a convergence of gene expression data on amine metabolism pathways following TGF-β-induced EMT, and validated upregulation of ammonia production using wet bench experimentation. Interestingly, we also identified ATG16L1, a regulator of autophagy, as a central node in an ammonia production gene network, suggesting connections between elevated amine metabolism, EMT, and autophagy. ATG16L1 was also found to be upregulated during TGF-β-induced EMT. Finally, using high- throughput drug screens, we showed that siRNA-mediated inhibition of the autophagy regulator, ATG16L1, rescued TGF-β-mediated chemoresistance. Together, this iterative combination of systems-based analyses and experimental validations suggests that TGF-β-mediated EMT converges on a gene expression network to induce autophagy and altered metabolism that can be targeted to overcome chemoresistance.

## 2. Materials and Methods

### 2.1. Cell Culture

All cell lines were obtained from the Duke Cell Culture Facility. The Duke Cell Culture Facility routinely tests for mycoplasma and performs cell line authentication by short tandem repeat analysis. Cells were cultured in Dulbecco’s Modified Eagle Medium (DMEM) with fetal bovine serum (FBS) and 1% penicillin–streptomycin in a standard 37 °C tissue culture incubator with 5% CO_2_. Cell confluence during vehicle and TGF-β treatment was measured using the IncuCyte Zoom live cell analysis system (Sartorius, Goettingen, Germany).

### 2.2. RNA Extraction, Reverse Transcription, and RT-qPCR

RNA extraction, reverse transcription, and RT-qPCR were performed as previously described [29].

### 2.3. Western Blotting

Cells were prepared and lysed in 1× radioimmunoprecipitation assay (RIPA) buffer (Thermo, catalog number: 89900, Waltham, MA, USA) mixed with 1× protease and phosphatase inhibitor cocktail (Roche, San Jose, CA, USA). The composition of the RIPA buffer was 25 mM Tris·HCl pH 7.6, 150 mM NaCl, 1% Nonidet P-40 (NP-40), 1% sodium deoxycholate, and 0.1% sodium dodecyl sulfonate (SDS). Cell lysates were incubated at 4 °C for 20 min. and centrifuged at 14,000*g* for 5 min. Cleared lysates were mixed with 4× Laemmli loading buffer and incubated at 95 °C for 3 min. Lysates were separated in 4–12% NuPAGE Novex Bis-Tris gels (ThermoFisher, Waltham, MA, USA). Proteins were transferred to nitrocellulose membrane (GE Healthcare Life Sciences, Pittsburgh, PA, USA) in 1× NuPAGE Transfer Buffer (ThermoFisher, Waltham, MA, USA) for 2 h at 75 V at 4 °C in the cold room. Membranes were blocked at room temperature using Starting Block T20 TBS Blocking Buffer (ThermoFisher, Waltham, MA, USA). Primary antibodies were added to the blocking buffer and incubated at 4 °C overnight. Membranes were washed two times for 5 min. each with phosphate-buffered saline (PBS) and incubated with LI-COR goat anti-mouse or goat anti-rabbit secondary antibodies diluted 1:20,000 in Starting Block buffer. Membranes were visualized using the Odyssey Fc imager (27402864, LI-COR Biosciences, Lincoln, NE, USA). Primary antibodies used included glyceraldehyde-phosphate dehydrogenase (GAPDH) (C2415, Santa Cruz Biotechtology; 1:1000, Dallas, TX, USA), ATG16L1 (8089T, Cell Signaling; 1:1000, Danvers, MA, USA) and LC3 A/B (12741T, Cell Signaling; 1:1000, Danvers, MA, USA).

### 2.4. Ammonia Production Assay

A total of 200,000 cells were seeded in 6 cm dishes. At each time point, cells were washed with PBS, scraped, and lysed in Ammonia Assay Buffer provided in the Abcam ammonia assay kit (ab83360, Abcam, Cambridge, UK) after the end of each treatment time point. Ammonia production assays were performed after collecting all time points using the protocol recommended by the manufacturer.

### 2.5. Cytoscape Analysis

Gene networks were analyzed by importing all available human data on each gene in the Universal Interaction Database Client using Cytoscape version 3.5.1 (https://cytoscape.org/). All relevant networks of genes were merged to visualize interactions among genes.

### 2.6. Phylogenetic Reconstructions from Gene Expression Data

Distance-based dendograms were created by first constructing a distance matrix calculated based on the entire microarray dataset for each dataset to be analyzed, using the genes as the characters, the raw expression value for each gene as the set of character states, and the samples as the taxa. The Neighbor Joining method [30] was used for reconstructing phylogeny with distance matrices. To perform analysis based on maximum-likelihood (ML) and parsimony, the continuous gene expression data was converted into categorical variables. For example, for GSE23038, we used the passage 0 sample as an ‘outgroup’, and converted the gene expression data for all other samples into either upregulated, downregulated, or constant relative to passage 0. The reliability of the parsimony method is generally considered to increase with an increasing number of informative characters [31,32,33]. Therefore, cut-off thresholds of up- and downregulation were determined by calculating the maximum number of informative sites given different cut-offs, and a threshold was selected that provided the highest number of informative sites in each dataset. ML and parsimony analyses were then performed based on converted data. ML analysis after data conversion was performed online on a free phylogeny platform, PhyML 3.0 (14), whereas distance and parsimony tree constructions were performed using the APE [34] and Phangorn [35] packages implemented in R (15). Bootstrap tests of 100 pseudoreplicates were performed for all phylogenies to assess the branch support. Tree files were visualized in FigTree (Andrew Rambaut; http://tree.bio.ed.ac.uk/software/figtree/).

### 2.7. High-Throughput Screening

A549 cells were screened with the NCI Approved Oncology Drugs Set VI in the presence of vehicle (4 mM HCl and 2% bovine serum albumin (BSA)) or 4 ng/mL recombinant human TGF-β (R&D Systems, Minneapolis, MN, USA). Briefly, A549 cells were dispensed using liquid handling into 384 well plates with no drug, dimethyl sulfoxide (DMSO), or 1 µM drug at cell plating densities of 250 or 1000 cells/well. Plates were incubated at 37 °C, and cell viability was assayed by CellTiterGlo (Promega, Madison, WI, USA) after 72 h. Relative drug resistance or sensitivity was calculated as the fold change difference in CellTiterGlo value between vehicle-treated and TGF-β-treated wells. To perform the screen in the context of ATG16L1 knockdown, 20 nM siRNA targeting ATG16L1 was delivered to A549 cells by reverse transfection using RNAiMax and incubated at 37 °C for 24 h. After 24 h, the drug screen was performed −/+ TGF-β, as described above. All screens were performed in the Duke Functional Genomics Shared Resource. Raw data for the screens are provided in Appendix A.

### 2.8. Correlation of ATG16L1 with Clinical Outcomes

Kaplan–Meier curves were generated based on patients stratified by ATG16L1 expression level using R2: Genomics Analysis and Visualization Platform (https://hgserver1.amc.nl/cgi-bin/r2/main.cgi) and GEPIA (http://gepia.cancer-pku.cn/). The scan option was used to automatically select the cut-off values in the R2 platform, and default settings were used for GEPIA.

### 2.9. Statistical Analyses

All assays were performed in triplicate, and all experiments were repeated a minimum of two times. The real-time polymerase chain reaction (qRT-PCR) and ammonia production assays were analyzed using a one-way ANOVA with Tukey’s post-hoc correction for multiple comparisons in JMP14.0 (SAS, Cary, NC, USA). Drug screen data was analyzed by linear regression and analysis of variance in JMP14.0. Pathway analyses were performed in FuncAssociate (http://llama.mshri.on.ca/funcassociate/) [36], which calculates an adjusted *p*-value as a fraction of 1000 simulations having attributes with the single hypothesis *p*-value. For Kaplan–Meier survival curves were analyzed by log-rank tests. All *p*-values <0.05 were considered statistically significant.

## 3. Results

### 3.1. Phylogenetics Analyses Provide a Simple and Reliable Tool to Visualize Gene Expression Dynamics

To test the feasibility and effectiveness of using phylogenetics as a clustering tool to analyze gene expression data, we tested if phylogenetic trees could recapitulate the temporal order of gene expression data collected at different time points. To do this, we constructed dendograms from publicly-available microarray data for immortalized prostate cells collected every 10 passages from 0 to 80 passages (GSE23038, [37]).

We first used distance-based trees to infer temporal relationships among the samples. Distance-based trees use a data matrix comprised of gene expression values as a continuous variable without the need for binning gene expression data into categorical variables of being upregulated, unchanged, and downregulated. Distance-based construction of a rooted tree with root at passage 0 produced a tree topology that, with the exception of passage 70, clustered samples according to their temporal order from passage 10 to 80 (Figure 2A).

We also analyzed GSE23038 [37] using maximum-likelihood and parsimony phylogenetics methods. The raw data matrix was converted into three character states based on a neutral evolution model, JC69, before being used as input for these two methods of tree construction. Importantly, for all three methods, trees constructed using gene expression data recapitulated the known temporal structure of the data with robust bootstrap support (Figure 2A–C, bootstrap values indicated above branches). A comparison of the three cladistical methods with clustering revealed that hierarchical clustering was unable to accurately reconstruct the temporal order of passages (Figure 2D,E).

Similarly, we performed phylogenetic clustering on additional datasets where samples had been analyzed longitudinally, including GSE17708 [38], microarray data from A549 lung adenocarcinoma cells treated with TGF-β over a period of 72 h, and GSE12548, microarray data from human ARPE-19 retinal pigment epithelium cells treated with TGF-β and TNF-α over 60 h [39]. For both of these datasets, phylogenetic clustering reconstructed the temporal order of treatments with strong bootstrap support (Figure 3A,B).

### 3.2. Analyzing Dynamics of TGF-β Treatment through Visualization of Tree Structure Reveals Two Distinct Temporally-Resolved Clades

A major advantage of clustering is its ability to easily visualize relationships between large datasets and to derive novel insights. For example, re-analysis of microarray data from A549 cells treated with TGF-β over 72 h (GSE17708) revealed two distinctive patterns in the resulting phylogenies. First, early time points (0–8 h) were haphazardly organized in clades and subclades, where replicates of samples were admixed, indicating that phylogenetic analyses were not able to provide a clear signal based on the expression data that would predict timing of treatment (Figure 4A). Second, the later time points (≥8 h) were well resolved, suggesting the presence of a clear signal emerging in the gene expression data following long-term treatment with TGF-β (Figure 4A).

Consistent with a convergence of signal at later time points, RT-qPCR analysis of the epithelial marker, E-cadherin, and the mesenchymal marker, vimentin, demonstrated that E-cadherin suppression and vimentin activation were not apparent until this bifurcation of early admixed time points vs. resolved late time points (Figure 4B). Likewise, our time-lapse imaging analysis of growth rate between vehicle-treated and TGF-β-treated A549 cells showed that differences in growth rate between the two conditions were not observed until approximately 72 h after the initiation of treatment (Figure 4C; Appendix A), consistent with reports demonstrating that EMT induces cell cycle arrest [40,41]. These experimental results suggest that the timing of both gene expression and phenotypic traits associated with EMT are consistent with the convergence of an emerging signal at later time points within the dendograms.

Next, we extracted genes that were differentially expressed across the two major clades of early and late treatment times. Pathway analysis of these genes showed that multiple amine metabolism pathways were significantly altered during TGF-β treatment (Figure 4D). To experimentally test if ammonia metabolism was altered during TGF-β treatment, we performed ammonia production assays on A549 cells from which the publicly available data were originally generated. Importantly, we found that ammonia production was altered significantly upon TGF-β treatment at later time points, with little change in ammonia production during earlier time points (Figure 4E). Together, these analyses demonstrated the utility of simple visualizations, such as phylogenetic trees and clustering dendograms, to yield new testable hypotheses.

### 3.3. Gene Expression Networks Couple Ammonia Production to Autophagy

Previous research has identified a connection between upregulation of ammonia production and induction of autophagy [42]. Based on this connection, we tested if TGF-β-induced EMT led to an increase in autophagy markers. In support of this hypothesis, TGF-β treatment led to upregulation of autophagy markers LC3A/B and ATG16L1 (Figure 5A,B). To better understand the connections between ammonia production and autophagy, we used Cytoscape to construct gene regulatory networks related to amine metabolism genes and autophagy regulators. We constructed gene networks that included the ammonia production genes identified by the pathway analysis, along with the autophagy markers LC3A/B and ATG16L1, that we identified in our western blots to be activated upon TGF-β treatment. Although we found few gene–gene interactions among amine metabolism genes alone (Figure 5C), when we added the autophagy regulator ATG16L1 to this network, it connected the entire set of previously-isolated amine metabolism subnetworks (Figure 5D). Our results suggest that TGF-β-mediated EMT is associated with increased amine production and upregulation of autophagy. It remains to be tested in this system if the ammonia production induces autophagy, as has been demonstrated previously in both yeast and mouse embryonic fibroblasts [42], or if TGF-β-induced autophagy upregulation leads to more ammonia. However, our results demonstrate a connection between TGF-β-mediated EMT, altered amine production, and upregulation of autophagy.

### 3.4. Autophagy Inhibition Re-sensitizes Cells to TGF-β-Induced Chemoresistance

Our data revealed that TGF-β-induced EMT leads to ammonia production and upregulation of autophagy. Interestingly, both EMT and autophagy are known to be involved in chemoresistance. EMT can drive chemoresistance in multiple cancers [43,44,45,46]. Likewise, autophagy is a pro-survival mechanism in response to cellular stresses, such as hypoxia and nutrient deprivation, and is increasingly implicated in resistance to cancer treatments [47,48]. Integrating our observations with these reports, we hypothesized that EMT-induced drug resistance is mediated, at least in part, by elevated levels of the autophagy regulator, ATG16L1.

To test this hypothesis, we used high-throughput drug screens of 119 FDA-approved small-molecule anticancer agents. To do this, we first tested if TGF-β-mediated EMT led to chemoresistance. We screened A549 cells treated with either vehicle or TGF-β and plated at both low and high density. After 72 h of incubation with each drug, the overall cell viability was analyzed with CellTiterGlo. We first performed quality control analyses of the screens. Linear regression of the empty and DMSO-treated wells showed virtually no relationship between the CellTiterGlo value and the position on the plate when comparing the same plate setup across multiple plates (*R*^2^ = 0.0862), suggesting that the screen results did not suffer from plate effects (Appendix A). By contrast, the correlation coefficients in drug-containing wells were greater than 0.8 between high and low cell density for both vehicle- and TGF-β-treated conditions, suggesting high reproducibility across replicate plates when drug is present in the well (Appendix A).

Given the lack of apparent plate effects and strong reproducibility between replicate screens, we investigated whether TGF-β induced chemoresistance. Consistent with our hypothesis, TGF-β treatment increased resistance to 60% (71/119) of the compounds tested, as evaluated by an increase in CellTiterGlo absorbance as compared to vehicle-treated control wells (Figure 6A). Analysis of these compounds by pathway targets showed that TGF-β induced resistance to both broad spectrum chemotherapies, such as microtubule-targeting agents and topoisomerase inhibitors, as well as multiple targeted therapies, including those against HER2 and EGFR (Figure 6B).

Next, to investigate the importance of autophagy in promoting TGF-β-induced therapy resistance, we performed siRNA-mediated knockdown of ATG16L1, the autophagy marker we identified as upregulated in TGF-β-treated cells. We first tested knockdown efficiency using four independent siRNAs and selected by Western blot analysis siRNA_1 for subsequent drug screens (Figure 6C). We then screened A549 with the same 119 drugs +/– TGF-β and treated with either a non-silencing siRNA or siRNA_1 targeting ATG16L1. Remarkably, ATG16L1 knockdown re-sensitized cells to 29/71 (41%) of drugs for which TGF-β treatment led to increased resistance (Figure 6D). Interestingly, these drugs included current standard-of-care therapies for small-cell lung cancer (SCLC), doxorubicin, and topotecan, as well as anti-VEGFR therapies, regorafenib, and axitinib, both of which have shown promising clinical benefits in early stage clinical trials against advanced non-small-cell lung cancer (NSCLC) [49,50], and cabozantinib, a tyrosine kinase inhibitor that has shown efficacy along with, or in combination with, erlotinib in the treatment of EGFR wild-type NSCLC patients [51]. Analysis by pathways showed that, on average, autophagy inhibition re-sensitized cells to multiple targeted therapies, including c-MET, c-RET, FLT3, TAM2, and dihydrofolate reductase (DHFR) (Figure 6E). Together, our results support the hypothesis that TGF-β-mediated therapy resistance is driven, in part, by the autophagy regulator ATG16L1, suggesting the potential use of autophagy inhibitors as a concurrent or adjuvant therapy to counter resistance.

To determine if ATG16L1 was related to clinical outcomes, we analyzed ATG16L1 expression in gene expression datasets from patient tumors. Analysis of Kaplan–Meier curves showed that low ATG16L1 expression is prognostic for improved overall survival in patients with lung and clear cell renal cancer (Figure 7A–C) and improved relapse-free survival in patients with colorectal cancer (Figure 7D). Together, these analyses indicate ATG16L1 as an important prognostic marker of clinical response and cancer cell aggression.

## 4. Discussion

The progression of cancer from an indolent, slow-growing primary tumor to metastatic and therapy resistant disease is, at its foundation, an evolutionary process. Genetic and genomic dysregulation promotes heterogeneity in tumor cell populations [52], which provides raw materials for selection of the fittest cancer cells. During this process, mutations [53], epigenetic alterations [54], and gene expression changes [55] are selected that enable survival of individual cancer cells under the diverse environmental pressures not only within the tumor, but also during metastatic progression [56,57] and the emergence of therapy resistance [58]. 

Here, we combined methods rooted in evolutionary theory, such as phylogenetic inference, with pathway and network analyses, as well as experimental techniques, to yield new insights. By taking this novel approach to analyze a well-established system—TGF-β-induced EMT—we identified mechanisms of therapy resistance. Specifically, we found that EMT leads to increased production of intracellular ammonia. Ammonia is a by-product of protein breakdown and serves an important function in maintaining homeostasis in electrolyte concentration [59]. Recent evidence, however, also suggests that ammonia production is involved in regulating autophagy and pro-survival circuits that contribute to chemoresistance [42,60]. Importantly, autophagy can lead to increased aggressiveness in cancer, perhaps as an adaptive response to cellular stress. In our present study, downregulation of the autophagy regulator, ATG16L1, partially reversed EMT-induced therapy resistance, suggesting the potential benefits of concurrent uses of autophagy inhibitors with standard-of-care therapies.

TGF-β has also been reported to induce metabolic reprogramming of stromal cells, such as cancer-associated fibroblasts (CAFs), where CAFs overexpressing TGF-β ligands show increased autophagy and HIF-1α activation and concomitantly reduced oxidative phosphorylation [61]. The scaffolding/regulatory protein caveolin-1—a functional regulator of TGF-β signaling—can play a key role in coordinating these responses [62,63]. Thus, the nexus of TGF-β signaling, increased autophagy, and metabolic reprogramming may be a common design principle of multiple cell types.

Interestingly, inhibition of autophagy consistently led to re-sensitization to c-Met inhibitors during EMT. The c-Met oncogene is one of the two most highly-mutated tyrosine kinase receptors in NSCLC, and resistance to tyrosine kinase inhibitors (TKI) invariably follows after treatment [64]. Indeed, resistance to erlotinib is common in lung cancer, and ATG16L1 knockdown re-sensitized cells to increased EMT-induced erlotinib resistance. EMT has been shown as an important contributor to this resistance as TKI resistance NSCLC cell lines has a more mesenchymal phenotype, higher expression of mesenchymal markers, such as Zeb-1 and vimentin, and downregulation of E-cadherin [65]. Recent evidence has shown that c-Met promotes anoikis resistance and cell growth via activation of autophagy regulators, such as ATG5 and Beclin-1 [66]. These observations suggest that autophagy may be an important resistance mechanism and a combinatorial use of autophagy inhibitors with TKIs may increase therapeutic efficacy of TKIs and possibly prolong or reverse resistance.

## 5. Conclusions

By integrating systems biology and experimental methodologies we have revealed new connections between EMT, autophagy, ammonia production, and chemoresistance. These studies demonstrate the power of coupling tools from evolutionary biology with systems-level informatics analysis and experimental validation to yield novel insights. Future work is aimed at better understanding the mechanistic connections between autophagy, ammonia production, and EMT to design new therapies to treat chemo-resistant disease. 

## Figures and Tables

**Figure 1 jcm-08-00205-f001:**
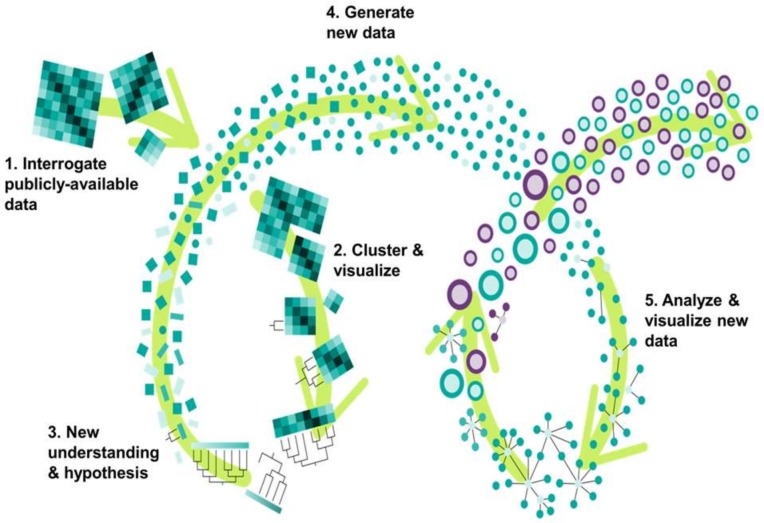
An integrated framework of iterative, systems-level analysis and experimental validation provides new insights. Large amounts of raw data, generated by new experimentation or re-analyzed from public databases (1), are analyzed by clustering approaches to easily visualize data topology (2). This visualization fosters a new, deeper understanding that informs a new hypothesis (3). Experimental validation of the new hypothesis generates new data (4), which is analyzed and visualized as a system (5).

**Figure 2 jcm-08-00205-f002:**
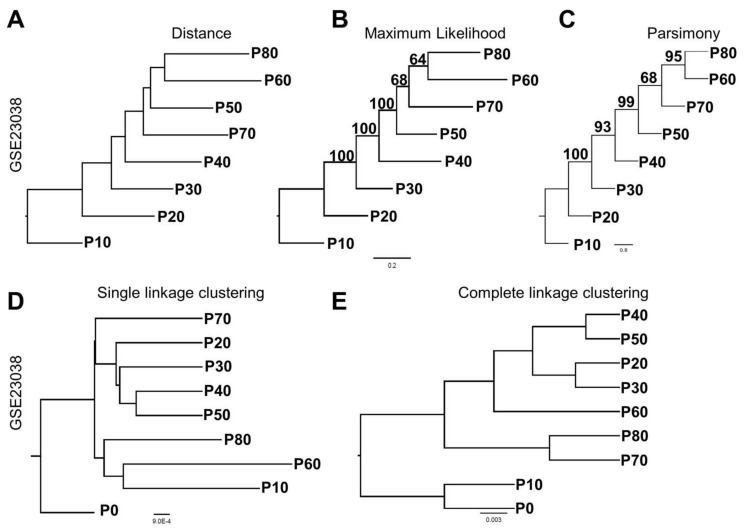
Phylogenetic reconstruction provides a simple visualization tool to view temporal changes in gene expression data. (**A**) Distance-based phylogeny of GSE23038; serial passage of normal prostate cells immortalized with hTERT using gene expression data as a continuous variable. (**B**) Maximum-likelihood and (**C**) maximum parsimony trees constructed based on gene expression data transformed to categorical variables. (**D**) Single and (**E**) complete linkage hierarchical clustering provides similar groupings of passage numbers, but lacks the temporal structure.

**Figure 3 jcm-08-00205-f003:**
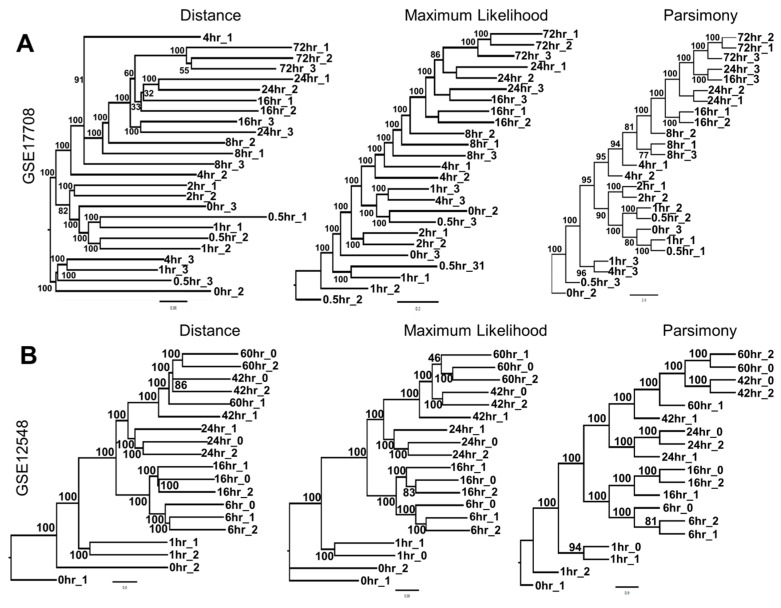
Phylogenetic clustering enables reconstruction of longitudinal data based on gene expression. (**A**) Distance, maximum parsimony, and maximum-likelihood dendograms of GSE17708; microarray analysis of A549 cells treated with TGF-β over 72 h. (**B**) Distance, maximum parsimony, and maximum-likelihood phylogeny construction of GSE12548; TGF-β and TNF-α treatment of human retinal pigment epithelium cells over 60 h.

**Figure 4 jcm-08-00205-f004:**
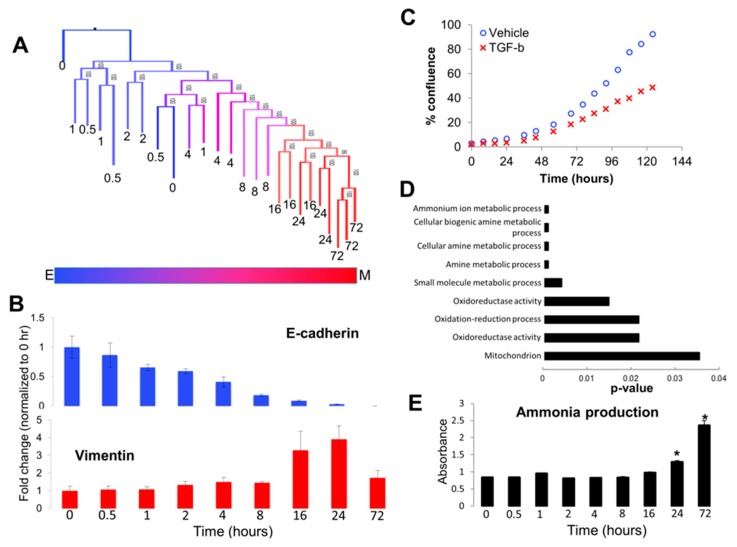
Visualization of tree topology reveals altered metabolism during epithelial–mesenchymal transition (EMT). (**A**) The topology of the maximum-likelihood reconstruction of GSE17708 showed an admixed clade at early time points in A549 cells with TGF-β treatment, with a clearly-resolved clade of later time points after eight hours as the phenotypic signal switched from epithelial to mesenchymal. (**B**) Consistent with the tree topology, changes in EMT biomarkers E-cadherin and vimentin were not apparent until after eight hours of treatment. * indicate *p* < 0.05 as compared to the 0 h time point (**C**) Growth curves of A549 cells treated with vehicle (blue circles) or TGF-β (red ×) analyzed by IncuCyte time-lapse imaging revealed TGF-β-induced growth inhibition at 48–72 h. (**D**) Pathway analysis of genes contributing to the bifurcation of early (<8 h) and late (≥8 h) time point clades revealed TGF-β-induced changes in amine metabolism pathways at the later time points as compared to the early time points. (**E**) Ammonia production assays validated the prediction that TGF-β induces upregulation of ammonia production.

**Figure 5 jcm-08-00205-f005:**
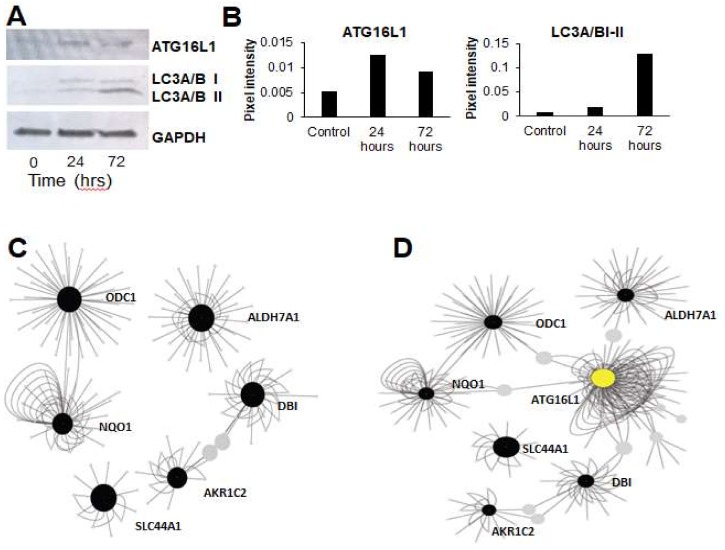
Epithelial-mesenchymal transition induces activation of autophagy and links to an amine production gene network. (**A**) TGF-β-induced epithelial-mesenchymal transition led to up-regulation of autophagy markers ATG16L1 and MAP1LC3A (LC3A/B). (**B**) Densitometric quantification of the western blotting data in A. (**C**) Cytoscape networks of amine production genes identified in Figure 4 showed few interactions between sub-networks. (**D**) Addition of the autophagy regulator, ATG16L1 (yellow circle), acted as a central hub to connect all amine metabolism sub-networks.

**Figure 6 jcm-08-00205-f006:**
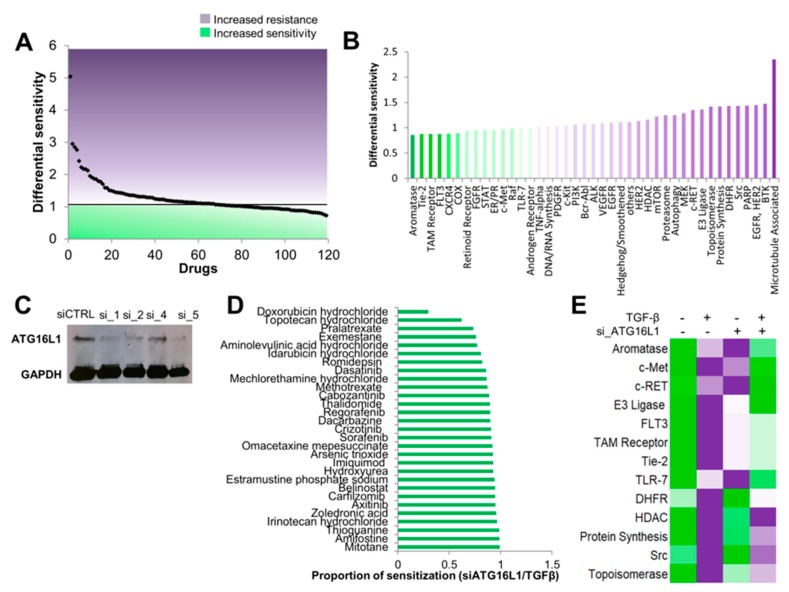
ATG16L1 knockdown rescues TGF-β-mediated chemoresistance. (**A**) A screen of 119 FDA-approved small molecule inhibitors demonstrated a broad increase in chemoresistance following TGF-β treatment. Each black dot represents one compound. Dots above the 1 were differentially resistant in TGF-β-treated cells as compared to vehicle-treated cells; dots below the 1 were more sensitive in the TGF-β-treated cells as compared to vehicle-treated cells. (**B**) Analysis of drug screen data by targets and pathways identified increased TGF-β-mediated resistance to several common chemotherapies, such as microtubule-associated and topoisomerase inhibitor therapies, and targeted therapies in lung cancer treatment, such as c-MET, VEGF, and EGFR (purple bars). (**C**) Knockdown of ATG16L1 by siRNAs was validated by western blotting. siCtrl = non-silencing siRNA; si_1, si_2, si_4, and si_5 are independent siRNAs targeting ATG16L1 (**D**) A549 lung adenocarcinoma cells −/+ TGF-β and −/+ siATG16_1 were screened against 119 FDA-approved compounds to identify drugs for which ATG16L1 rescued TGF-β-mediated therapy resistance. ATG16L1 knockdown re-sensitized cells to multiple therapeutic agents. (**E**) Pathway level analysis of compounds where TGF-β-mediated resistance was rescued by ATG16L1 knockdown.

**Figure 7 jcm-08-00205-f007:**
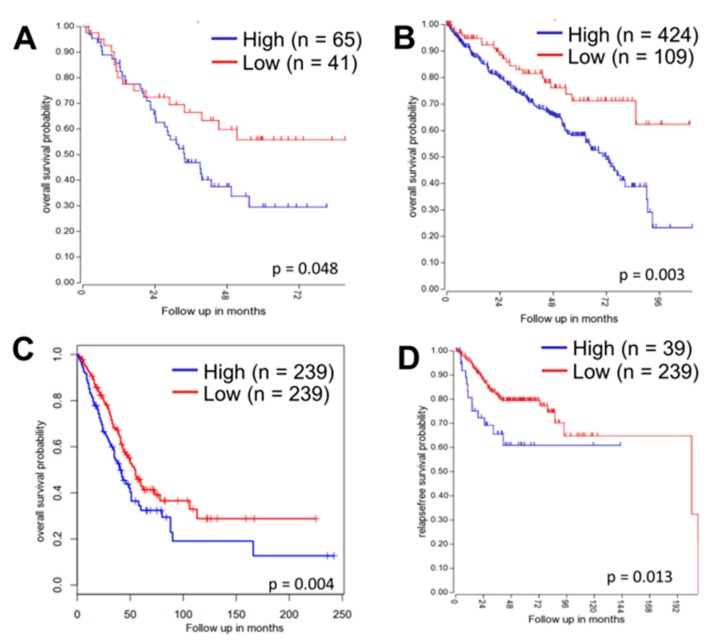
ATG16L1 is a prognostic biomarker of survival and progression in carcinoma patients. (**A**) Low ATG16L1 expression is prognostic for improved overall survival in lung adenocarcinoma patients. (**B**) Low ATG16L1 expression significantly predicts improved overall survival in kidney renal clear cell carcinoma patients. (**C**) Lower ATG16L1 expression in lung adenocarcinoma from The Cancer Genome Atlas dataset is prognostic for improved overall survival; data analyzed using GEPIA—http://gepia.cancer-pku.cn/. (**D**) Low ATG16L1 expression trends with better relapse-free survival in colorectal carcinoma patients.

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
