# Peer review of "An Integrative Systems Biology and Experimental Approach Identifies Convergence of Epithelial Plasticity, Metabolism, and Autophagy to Promote Chemoresistance"

_jcm, 2019, doi:10.3390/jcm8020205_

Reviewer 1 Report

Authors have used TGF-b/SMAD axis, a well-studied signaling pathway implicated in cancer and a known player in EMT, to demonstrate the usefulness of combining evolutionary and systems biology with experimental validation to delineate a new system to target key proteins to overcome chemoresistance. This study is quite relevant to the current trends of bioinformatics and experimental approach, however a few minor points will elevate the quality of work and enhance rigor of the study.

There is no explanation as to how the statistics was performed and what was considered to be significant. 

Number of biological and technical replicates of treatments and assays performed (Western blots, qPCR etc) need to be mentioned in the Materials and Methods section.

Image of the cells in culture for IncuCyte time lapse study (figure 4C) would be a very meaningful addition to the article as that will give a visual confirmation of the cell morphology.

Western blot images (figure 5A, 5C) are a little difficult to comprehend. Densitometric analysis of the band intensities will help with the conclusions in the article.

Explanation as to how cell confluence was measured is highly recommended.

In addition to cell viability, cleaved caspase or Annexin PI for measuring apoptosis would be beneficial for the paper.

Author Response

Comment 1: “There is no explanation as to how the statistics was performed and what was considered to be significant.”

Response 1: We thank the reviewer for pointing this out. We have included a section “Statistical analysis” in the materials and methods.

Comment 2: “Number of biological and technical replicates of treatments and assays performed (Western blots, qPCR etc) need to be mentioned in the Materials and Methods section.”

Response 2: We agree. We have included this information in the “Statistical analysis” section in the materials and methods.

Comment 3: “Image of the cells in culture for IncuCyte time lapse study (figure 4C) would be a very meaningful addition to the article as that will give a visual confirmation of the cell morphology.”

Response 3: We have included images from the IncuCyte imaging of TGF-β treatment in Figure 4C. We have also included videos of A549 cells treated with vehicle or TGF-β in supplementary materials.

Comment 4: “Western blot images (figure 5A, 5C) are a little difficult to comprehend. Densitometric analysis of the band intensities will help with the conclusions in the article.”

Response 4: We have updated the figure to include a densitometric analysis of the band intensities of the western blots.

Comment 5: “Explanation as to how cell confluence was measured is highly recommended.”

Response 5: We agree. We have included an explanation of how cell confluence was measured.

Comment 6: “In addition to cell viability, cleaved caspase or Annexin PI for measuring apoptosis would be beneficial for the paper.”

Response 6: The treatment of A549 cells with TGF-b is a classic and well-known model system to study epithelial-mesenchymal transition. Indeed, the relationship between TGF-b and apoptosis has been thoroughly characterized in the literature. TGF-b has a dual role in mediating apoptosis of A549 cells in the presence of TGF-b. On one hand, Huang et al. 2000 (PMID: 10748131) showed TGF-b blocked apoptosis of A549 cells in response to serum deprivation. Conversely, Bai et al. (2011) showed that acute exposure of A549 cells to TGF-b led to Fas-induced apoptosis, but long-term exposure to TGF-b inhibited apoptosis (PMID: 21539941). Based on the extensive literature that already exists on this topic, we agreed the addition of apoptosis studies to our analysis would not add significantly to the conclusions of the manuscript.

Reviewer 2 Report

Xu et al. report results obtained by different combined experimental and network methods, with the goal to demonstrate a mechanism involved in chemoresistance, and by identifying in TGFβ-induced-EMT and cellular autophagy the main targets of the study.   

It is an interesting and a well conducted study.  So, this referee  suggests only some minor revisions, as stated below.

Minor Revisions:

1. EMT experiments were performed in A549 lung adenocarcinoma cells, which don’t represent the better cellular model in evaluating EMT event, such instead BEAS-2B cells, for example. This point has to be discussed by authors.

2. Since it has been stated by authors (page 7, line 235), the phenotypic change after TGF-β treatment, typical of EMT event, should be documented by authors (by a photo, for example). This referee thinks it should be better if authors provide this evidence.

3. Figure 5, panel A: the figure has a poor resolution, authors should improve the image. Moreover, authors have to provide the densitometric analysis of the western blot.

Author Response

Reviewer 2

Comment 1: “EMT experiments were performed in A549 lung adenocarcinoma cells, which don’t represent the better cellular model in evaluating EMT event, such instead BEAS-2B cells, for example. This point has to be discussed by authors.”

Response 1: A549 cells are a standard cell line for mechanistic studies of TGF-β treatment. Indeed, a search of the literature in Pubmed for “A549” and “TGF” produces 617 articles while the same search for “BEAS-2B” and “TGF” produces just 81 articles. Perhaps more importantly, the publicly-available data we analyzed was from A549 cells, which prompted us to use A549 cells to validate our findings. We have included a discussion of the use of A549 cells in the text to highlight that we wished to validate the findings in the same cell line from which the gene expression data was derived (line 202-203, Results section).

Comment 2: “Since it has been stated by authors (page 7, line 235), the phenotypic change after TGF-β treatment, typical of EMT event, should be documented by authors (by a photo, for example). This referee thinks it should be better if authors provide this evidence.”

Response 2: We agree. We have included images showing vehicle and TGF-b treatment of A549 cells in Figure 4C along with supplementary movies showing the growth and morphological changes during the treatment time.

Comment 3: Figure 5, panel A: the figure has a poor resolution, authors should improve the image. Moreover, authors have to provide the densitometric analysis of the western blot.

Response 3: We have provided an improved image and included the densitometric analysis as requested.

Reviewer 3 Report

Xu et al use an elegant mix of systems and experimental approaches to show convergence of epithelial-mesenchymal plasticity, metabolic switching and autophagy to promote chemoresistance. Overall, I am enthusiastic about this work. However, I recommend addressing a few major points, all related to statistics and replicates, and a few minor points.

Major comments

1. Where p values are reported, please specify which statistical tests were used, e.g. Figure 4B-E, Supplementary Figures 1 (also add p values please), Figure 7A-D.

2. Please indicate how many replicates were used in Figures 4C and 6A. The curves in Figure 4C look consistent, but for Figure 6, if only one replicate was used, how do the authors ensure that individual results are reproducible and are not ‘outliers’ if only one replicate with only one hairpin was used?

3. Line 339, statement “ATG16L1 as an important prognostic marker” and Survival curves in Figure 7: I assume that p-values are from log-ranks tests? Are survival differences still significantly different in multivariable analyses (e.g. using Cox regression) including clinical parameters like age at diagnosis and stage?

Minor comments

- Page 3, Line 108: RIPA buffer: please include exact composition, as this may vary from lab to lab.

- Page 3, Line 120: Were all primary antibody dilutions used at 1:1000?

- Figures 2-7: Please remove the labels “Figure 2”, “Figure 3”, etc. at the top-left of each.

- Page 5, lines 180-181: Duplicate sentence.

- Pages 12,13: Please remove duplicate Figure legends, already included above.

 Author Response

Reviewer 3

Comment 1: Where p values are reported, please specify which statistical tests were used, e.g. Figure 4B-E, Supplementary Figures 1 (also add p values please), Figure 7A-D.

Response 1: We thank the reviewer for pointing this out. We have included a section on “Statistical analyses” in the materials and methods and added p-values to Supplementary Figure 1.

Comment 2: Please indicate how many replicates were used in Figures 4C and 6A. The curves in Figure 4C look consistent, but for Figure 6, if only one replicate was used, how do the authors ensure that individual results are reproducible and are not ‘outliers’ if only one replicate with only one hairpin was used?

Response 2: We have included a section in the materials and methods to describe the number of biological replicates and experimental replicates performed for all assays.

Comment 3: Line 339, statement “ATG16L1 as an important prognostic marker” and Survival curves in Figure 7: I assume that p-values are from log-ranks tests? Are survival differences still significantly different in multivariable analyses (e.g. using Cox regression) including clinical parameters like age at diagnosis and stage?

Response 3: The reviewer is correct that all p-values are from log rank tests. We have included a description of these analyses in the materials and methods section. We were unable to assess if these differences are significant using multivariate Cox regression analyses to include age at diagnosis or stage. While we agree these would be useful analyses, the R2 package we used for these analyses does not allow for these custom parameters to be included.

Comment 4: Page 3, Line 108: RIPA buffer: please include exact composition, as this may vary from lab to lab.

Response 4: We have included the catalog number and composition for the RIPA buffer used.

Comment 5: Page 3, Line 120: Were all primary antibody dilutions used at 1:1000?

Response 5: Yes. We have updated the materials and methods to specify the dilution for each of the primary antibodies used.

Comment 6: Figures 2-7: Please remove the labels “Figure 2”, “Figure 3”, etc. at the top-left of each.

Response 6: We have removed the figure number labels at the top of each figure.  

Comment 7: Page 5, lines 180-181: Duplicate sentence.

Response 7: We have re-read the entire manuscript carefully; however, we were unable to find the duplicate sentence on page 5 or lines 180-181.

Comment 8: Pages 12,13: Please remove duplicate Figure legends, already included above.

Response 8: We were unable to find duplicate Figure legends in the text.